# Order Optimal One-Shot Distributed Learning

**Arsalan Sharifnassab,   Saber Salehkaleybar,   S. Jamaloddin Golestani**

Department of Electrical Engineering, Sharif University of Technology, Tehran, Iran

a.sharifnassab@gmail.com,   saleh@sharif.edu,   golestani@sharif.edu

## Abstract

We consider distributed statistical optimization in one-shot setting, where there are $m$ machines each observing $n$ i.i.d. samples. Based on its observed samples, each machine then sends an $O(\log(mn))$-length message to a server, at which a parameter minimizing an expected loss is to be estimated. We propose an algorithm called *Multi-Resolution Estimator* (MRE) whose expected error is no larger than $\tilde{O}\big(m^{-1/\max(d,2)}n^{-1/2}\big)$, where $d$ is the dimension of the parameter space. This error bound meets existing lower bounds up to poly-logarithmic factors, and is thereby order optimal. The expected error of MRE, unlike existing algorithms, tends to zero as the number of machines ($m$) goes to infinity, even when the number of samples per machine ($n$) remains upper bounded by a constant. This property of the MRE algorithm makes it applicable in new machine learning paradigms where $m$ is much larger than $n$.

## 1   Introduction

The rapid growth in the size of datasets has given rise to distributed models for statistical learning, in which data is not stored on a single machine. In several recent learning applications, it is commonplace to distribute data across multiple machines, each of which processes its own data and communicates with other machines to carry out a learning task. The main bottleneck in such distributed settings is often the communication between machines, and several recent works have focused on designing communication-efficient algorithms for different machine learning applications [Duchi et al., 2012, Braverman et al., 2016, Chang et al., 2017, Diakonikolas et al., 2017, Lee et al., 2017].

In this paper, we consider the problem of statistical optimization in a distributed setting as follows. Consider an unknown distribution $P$ over a collection, $\mathcal{F}$, of differentiable convex functions with Lipschitz first order derivatives, defined on a convex region in $\mathbb{R}^d$. There are $m$ machines, each observing $n$ i.i.d sample functions from $P$. Each machine processes its observed data, and transmits a signal of certain length to a server. The server then collects all the signals and outputs an estimate of the parameter $\theta^*$ that minimizes the expected loss, i.e., $\min_\theta \mathbb{E}_{f\sim P}\big[f(\theta)\big]$. See Fig. 1 for an illustration of the system model.

We focus on the distributed aspect of the problem considering arbitrarily large number of machines ($m$) and

a) present an order optimal algorithm with $b = O(\log mn)$ bits per transmission, whose estimation error is no larger than $\tilde{O}\big(m^{-1/\max(d,2)}n^{-1/2}\big)$, meeting the lower bound in [Salehkaleybar et al., 2019] up to a poly-logarithmic factor (cf. Theorem 1);

b) we present an algorithm with a single bit per message with expected error no larger than $\tilde{O}\big(m^{-1/2} + n^{-1/2}\big)$ (cf. Proposition 1).

## 1.1 Background

The distributed setting considered here has recently employed in a new machine learning paradigm called *Federated Learning* [Konečnỳ et al., 2015]. In this framework, training data is kept in users' computing devices due to privacy concerns, and the users participate in the training process without revealing their data. As an example, Google has been working on this paradigm in their recent project, *Gboard* [McMahan and Ramage, 2017], the Google keyboard. Besides communication constraints, one of the main challenges in this paradigm is that each machine has a small amount of data. In other words, the system operates in a regime that $m$ is much larger than $n$ [Chen et al., 2017].

A large body of distributed statistical optimization/estimation literature considers "one-shot" setting, in which each machine communicates with the server merely once [Zhang et al., 2013]. In these works, the main objective is to minimize the number of transmitted bits, while keeping the estimation error as low as the error of a centralized estimator, in which the entire data is co-located in the server.

If we impose no limit on the communication budget, then each machine can encode its entire data into a single message and sent it to the server. In this case, the sever acquires the entire data from all machines, and the distributed problem reduces to a centralized problem. We call the sum of observed functions at all machines as the centralized empirical loss, and refer to its minimizer as the centralized solution. It is part of the folklore that the centralized solution is order optimal and its expected error is $\Theta\big(1/\sqrt{mn}\big)$ [Lehmann and Casella, 2006, Zhang et al., 2013]. Clearly, no algorithm can beat the performance of the best centralized estimator.

Zhang et al. [2012] studied a simple averaging method where each machine obtains the empirical minimizer of its observed functions and sends this minimizer to the server through an $O(\log mn)$ bit message. Output of the server is then the average of all received empirical minimizers. Zhang et al. [2012] showed that the expected error of this algorithm is no larger than $O\big(1/\sqrt{mn} + 1/n\big)$, provided that: 1- all functions are convex and twice differentiable with Lipschitz continuous second derivatives, and 2- the objective function $\mathbb{E}_{f\sim P}\big[f(\theta)\big]$ is strongly convex at $\theta^*$. Under the extra assumption that the functions are three times differentiable with Lipschitz continuous third derivatives, Zhang et al. [2012] also present a bootstrap method whose expected error is $O\big(1/\sqrt{mn} + 1/n^{1.5}\big)$. It is easy to see that, under the above assumptions, the averaging method and the bootstrap method achieve the performance of the centralized solution if $m \leq n$ and $m \leq n^2$, respectively. Recently, Jordan et al. [2018] proposed to optimize a surrogate loss function using Taylor series expansion. This expansion can be constructed at the server by communicating $O(m)$ number of $d$-dimensional vectors. Under similar assumption on the loss function as in [Zhang et al., 2012], they showed that the expected error of their method is no larger than $O\big(1/\sqrt{mn} + 1/n^{9/4}\big)$. It, therefore, achieves the performance of the centralized solution for $m \leq n^{3.5}$. However, note that when $n$ is fixed, all aforementioned bounds remain lower bounded by a positive constant, even when $m$ goes to infinity.

For the problem of sparse linear regression, Braverman et al. [2016] proved that any algorithm that achieves optimal minimax squared error, requires to communicate $\Omega(m \times \min(n, d))$ bits in total from machines to the server. Later, Lee et al. [2017] proposed an algorithm that achieves optimal mean squared error for the problem of sparse linear regression when $d < n$.

Recently, Salehkaleybar et al. [2019] studied the impact of communication constraints on the expected error, over a class of first order differentiable functions with Lipschitz continuous derivatives. In parts of their results, they showed that under the assumptions of Section 2 of this paper in the case of $\log mn$ bits communication budget, the expected error of any estimator is lower bounded by $\tilde{\Omega}\big(m^{-1/\max(d,2)}n^{-1/2}\big)$. They also showed that if the number of bits per message is bounded by a constant and $n$ is fixed, then the expected error remains lower bounded by a constant, even when the number of machines goes to infinity.

Other than one-shot communication, there is another major communication model that allows for several transmissions back and forth between the machines and the server. Most existing works of this type [Bottou, 2010, Lian et al., 2015, Zhang et al., 2015, McMahan et al., 2017] involve variants of stochastic gradient descent, in which the server queries at each iteration the gradient of empirical loss at certain points from the machines. The gradient vectors are then aggregated in the server to update the model's parameters. The expected error of such algorithms typically scales as $O\big(1/k\big)$, where $k$ is the number of iterations.

## 1.2 Our contributions

We study the problem of one-shot distributed learning under milder assumptions than previously available in the literature. We assume that loss functions, $f \in \mathcal{F}$, are convex and differentiable with Lipschitz continuous first order derivatives. This is in contrast to the works of [Zhang et al., 2012] and [Jordan et al., 2018] that assume Lipschitz continuity of second or third derivatives. The reader should have in mind this model differences, when comparing our bounds with the existing results.

Unlike existing works, our results concern the regime where the number of machines $m$ is large, and our bounds tend to zero as $m$ goes to infinity, even if the number of per-machine observations $n$ is bounded by a constant. This is contrary to the algorithms in [Zhang et al., 2012], whose errors tend to zero only when $n$ goes to infinity. In fact, when $n = 1$, a simple example[1] shows that the expected errors of the simple averaging and bootstrap algorithms in [Zhang et al., 2012] remain lower bounded by a constant, for all values of $m$. The algorithm in [Jordan et al., 2018] suffers from the same problem and its expected error may not go to zero when $n = 1$.

In this work, we present an algorithm with $O\big(\log(mn)\big)$ bits per message, which we call Multi-Resolution Estimator for Convex landscapes and $\log mn$ bits communication budget (MRE-C-log) algorithm. We show that the estimation error of MRE-C-log algorithm meets the aforementioned lower bound up to a poly-logarithmic factor. More specifically, we prove that the expected error of MRE-C-log algorithm is no larger than $O\big(m^{-1/\max(d,2)}n^{-1/2}\big)$. In this algorithm, each machines reports not only its empirical minimizer, but also some information about the derivative of its empirical loss at some randomly chosen point in a neighborhood of this minimizer. To provide insight into the underlying idea behind MRE-C-log algorithm, we also present a simple naive approach whose error tends to zero as the number of machines goes to infinity. Comparing with the lower bound in [Salehkaleybar et al., 2019], the expected error of MRE-C-log algorithm meets the lower bound up to a poly-logarithmic factor. Moreover, for the case of having constant bits per message, we present a simple algorithm whose error goes to zero with rate $\tilde{O}\big(m^{-1/2} + n^{-1/2}\big)$, when $m$ and $n$ go to infinity simultaneously. We evaluate performance of the MRE-C-log algorithm in two different machine learning tasks and compare with the existing methods in [Zhang et al., 2012]. We show via experiments, for the $n = 1$ regime, that MRE-C-log algorithm outperforms these algorithms. The observations are also in line with the expected error bounds we give in this paper and those previously available. In particular, in the $n = 1$ regime, the expected error of MRE-C-log algorithm goes to zero as the number of machines increases, while the expected errors of the previously available estimators remain lower bounded by a constant.

## 1.3 Outline

The paper is organized as follows. We begin with a detailed model and problem definition in Section 2. In Section 3, we present our algorithms and main upper bounds. We then report our numerical experiments in Section 4. Finally, in Section 5 we discuss our results and present open problems and directions for future research. The proofs of the main results and optimality of the MRE-C-log algorithm are given in the appendix.

## 2 Problem Definition

Consider a positive integer $d$ and a collection $\mathcal{F}$ of real-valued convex functions over $[-1, 1]^d$. Let $P$ be an unknown probability distribution over the functions in $\mathcal{F}$. Consider the expected loss function

$$F(\theta) = \mathbb{E}_{f \sim P}\big[f(\theta)\big], \qquad \theta \in [-1, 1]^d. \tag{1}$$

Our goal is to learn a parameter $\theta^*$ that minimizes $F$:

$$\theta^* = \operatorname*{argmin}_{\theta \in [-1,1]^d} F(\theta). \tag{2}$$

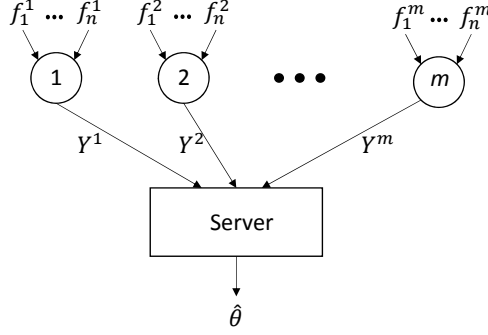

Figure 1: A distributed system of $m$ machines, each having access to $n$ independent sample functions from an unknown distribution $P$. Each machine sends a signal to a server based on its observations. The server receives all signals and output an estimate $\hat{\theta}$ for the optimization problem in (2).

The expected loss is to be minimized in a distributed fashion, as follows. We consider a distributed system comprising $m$ identical machines and a server. Each machine $i$ has access to a set of $n$ independently and identically distributed samples $\{f_1^i, \cdots, f_n^i\}$ drawn from the probability distribution $P$. Based on these observed functions, machine $i$ then sends a signal $Y^i$ to the server. We assume that the length of each signal is limited to $b$ bits. The server then collects signals $Y^1, \ldots, Y^m$ and outputs an estimation of $\theta^*$, which we denote by $\hat{\theta}$. See Fig. 1 for an illustration of the system model.[2]

**Assumption 1** *We let the following assumptions on $\mathcal{F}$ and $P$ be in effect throughout the paper.*

- *Every $f \in \mathcal{F}$ is once differentiable and convex.*

- *Each $f \in \mathcal{F}$ has bounded and Lipschitz continuous derivatives. More concretely, for any $f \in \mathcal{F}$ and any $\theta, \theta' \in [-1, 1]^d$, we have $|f(\theta)| \leq \sqrt{d}$, $\|\nabla f(\theta)\| \leq 1$, and $\|\nabla f(\theta) - \nabla f(\theta')\| \leq \|\theta - \theta'\|$.*

- *Distribution $P$ is such that $F$ (defined in (1)) is strongly convex. More specifically, there is a constant $\lambda > 0$ such that for any $\theta_1, \theta_2 \in [-1, 1]^d$, we have $F(\theta_2) \geq F(\theta_1) + \nabla F(\theta_1)^T (\theta_2 - \theta_1) + \lambda \|\theta_2 - \theta_1\|^2$.*

- *The minimizer of $F$ lies in the interior of the cube $[-1, 1]^d$. Equivalently, there exists $\theta^* \in (-1, 1)^d$ such that $\nabla F(\theta^*) = \mathbf{0}$.*

## 3    Algorithms and Main Results

In this section, we propose estimators to minimize the expected loss, organized in a sequence of three subsections. In the first subsection, we consider the case of constant bits per signal transmission, whereas in the last two subsections we allow for $\log mn$ bits per signal transmission. For the latter regime, we first present in Subsection 3.2, a simple naive approach whose estimation error goes to zero for large values of $m$, even when $n = 1$. Afterwards, in Subsection 3.3, we describe our main estimator, establish an upper bound on its estimation error, and show that it is order optimal.

### 3.1    Constant number of bits per transmission

Here, we consider a simple case with a one-dimensional domain ($d = 1$) and one-bit signal per transmission ($b = 1$). We show that the expected error can be made arbitrarily small as $m$ and $n$ go to infinity simultaneously.

**Proposition 1** *Suppose that $d = 1$ and $b = 1$. There exists a randomized estimator $\hat{\theta}$ such that*

$$\mathbb{E}\big[(\hat{\theta} - \theta^*)^2\big]^{1/2} = O\left(\frac{1}{\sqrt{n}} + \frac{1}{\sqrt{m}}\right).$$

The proof is given in Appendix A. There, we assume for simplicity that the domain is the $[0, 1]$ interval and propose a simple randomized algorithm in which each machine $i$ first computes an $O(1/\sqrt{n})$-accurate estimation $\theta^i$ based on its observed functions. It then sends a $Y^i = 1$ signal with probability $\theta^i$. The server then outputs the average of the received signals as the finial estimate.

Based on Proposition 1, there is an algorithm that achieves any desired accuracy even with budget of one bit, provided that $m$ and $n$ go to infinity simultaneously. In contrary, it was shown in Proposition 1 of [Salehkaleybar et al., 2019] that no estimator yields error better than a constant if $n = 1$ and the number of bits per transmission is a constant independent of $m$. We conjecture that the bound in Proposition 1 is tight. More concretely, for constant number of bits per transmission and any randomized estimator $\hat{\theta}$, we have $\mathbb{E}[(\hat{\theta} - \theta^*)^2]^{1/2} = \tilde{\Omega}\left(1/\sqrt{n} + 1/\sqrt{m}\right)$.

## 3.2    A simple naive approach with $\log mn$ bits per transmission

We now consider the case where the number of bits per transmission is $O(\log m)$. In order to set the stage for our main algorithm given in the next subsection, here we present a simple algorithm and show that its estimation error decays as $O(m^{-1/3})$. The underlying idea is that unlike existing estimators, in this algorithm each machine encodes in its signal some information about the shape of its observed functions at a point that is not necessarily close to its own private optimum. To simplify the presentation, here we confine our setting to one dimensional domain ($d = 1$) with each machine observing a single sample function ($n = 1$). The algorithm is as follows:

> Consider a regular grid of size $\sqrt[3]{m}/\log(m)$ over the $[-1, 1]$ interval. Each machine $i$ selects a grid point $\theta^i$ uniformly at random. The machine then forms a signal comprising two parts: 1- The location of $\theta^i$, and 2- The derivative of its observed function $f^i$ at $\theta^i$. In other words, the signal $Y^i$ of the $i$-th machine is an ordered pair of the form $\left(\theta^i, f'^i(\theta^i)\right)$, where $f'^i(\theta^i)$ is the derivative of $f^i$ at $\theta^i$. In this encoding, we use $O(\log m)$ bits to represent both $\theta^i$ and $f'^i(\theta^i)$. In the server, for each grid point $\theta$, the average of $f'^i$ is computed over all machines $i$ with $\theta^i = \theta$. We denote this average by $\hat{F}'(\theta)$. The server then outputs a point $\theta$ that minimizes $\left|\hat{F}'(\theta)\right|$.

This algorithm learns an estimation of derivatives of $F$, and finds a point that minimizes the size of this derivative. The following lemma shows that the estimation error of this algorithm is $\tilde{O}(1/\sqrt[3]{m})$. The proof is given in Appendix B.

**Proposition 2** *Let $\hat{\theta}$ be the output of the above estimator. For any $\alpha > 1$,*

$$\Pr\left(|\hat{\theta} - \theta^*| > \frac{3\alpha \log(m)}{\lambda \sqrt[3]{m}}\right) = O\left(\exp\left(-\alpha^2 \log^3 m\right)\right).$$

*Consequently, for any $k \geq 1$, we have $\mathbb{E}\left[|\hat{\theta} - \theta^*|^k\right] = O\left((\log(m)/\sqrt[3]{m})^k\right)$.*

We now turn to the general case with arbitrary values for $d$ and $n$, and present our main estimator.

## 3.3    The Main Algorithm

In this part, we propose our main algorithm and an upper bound on its estimation error. In the proposed algorithm, transmitted signals are designed such that the server can construct a multi-resolution view of gradient of function $F(\theta)$ around a promising grid point. Then, we call the proposed algorithm "Multi-Resolution Estimator for Convex landscapes with $\log mn$ bits communication budget (MRE-C-log)". The description of MRE-C-log is as follows:

Each machine $i$ observes $n$ functions and sends a signal $Y^i$ comprising three parts of the form $(s, p, \Delta)$. The signals are of length $O(\log(mn))$ bits and the three parts $s$, $p$, and $\Delta$ are as follows.

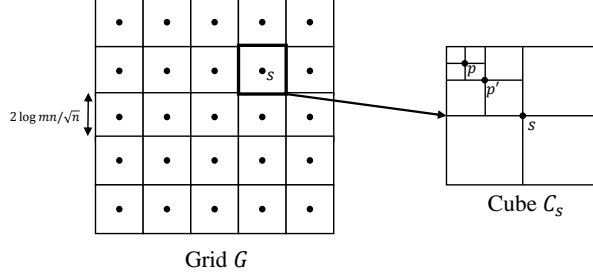

Grid $G$

Cube $C_s$

Figure 2: An illustration of grid $G$ and cube $C_s$ centered at point $s$ for $d = 2$. The point $p$ belongs to $\tilde{G}_s^2$ and $p'$ is the parent of $p$.

- Part $s$: Consider a grid $G$ with resolution $\log(mn)/\sqrt{n}$ over the $d$-dimensional cube. Each machine $i$ computes the minimizer of the average of its first $n/2$ observed functions,

$$\theta^i = \operatorname*{argmin}_{\theta \in [-1,1]^d} \sum_{j=1}^{n/2} f_j^i(\theta). \tag{3}$$

It then lets $s$ be the closest grid point to $\theta^i$.

- Part $p$: Let

$$\delta \triangleq 4\sqrt{d} \left( \frac{\log^5(mn)}{m} \right)^{\frac{1}{\max(d,2)}}. \tag{4}$$

Note that $\delta = \tilde{O}\big(m^{-1/\max(d,2)}\big)$. Let $t = \log(1/\delta)$. Without loss of generality we assume that $t$ is an integer. Let $C_s$ be a $d$-dimensional cube with edge size $2\log(mn)/\sqrt{n}$ centered at $s$. Consider a sequence of $t+1$ grids on $C_s$ as follows. For each $l = 0, \ldots, t$, we partition the cube $C_s$ into $2^{ld}$ smaller equal sub-cubes with edge size $2^{-l+1}\log(mn)/\sqrt{n}$. The $l$th grid $\tilde{G}_s^l$ comprises the centers of these smaller cubes. Then, each $\tilde{G}_s^l$ has $2^{ld}$ grid points. For any point $p'$ in $\tilde{G}_s^l$, we say that $p'$ is the parent of all $2^d$ points in $\tilde{G}_s^{l+1}$ that are in the $\big(2^{-l} \times (2\log mn)/\sqrt{n}\big)$-cube centered at $p'$ (see Fig. 2). Thus, each point $\tilde{G}_s^l$ ($l < t$) has $2^d$ children.

To select $p$, we randomly choose an $l$ from $0, \ldots, t$ with probability $2^{(d-2)l}/(\sum_{j=0}^{t} 2^{(d-2)j})$. We then let $p$ be a uniformly chosen random grid point in $\tilde{G}_s^l$. Note that $O(d\log(1/\delta)) = O(d\log(mn))$ bits suffice to identify $p$ uniquely.

- Part $\Delta$: We let

$$\hat{F}^i(\theta) \triangleq \frac{2}{n} \sum_{j=n/2+1}^{n} f_j^i(\theta), \tag{5}$$

and refer to it as the empirical function of the $i$th machine. If the selected $p$ in the previous part is in $\tilde{G}_s^0$, i.e., $p = s$, then we set $\Delta$ to the gradient of $\hat{F}^i$ at $\theta = s$. Otherwise, if $p$ is in $\tilde{G}_s^l$ for $l \geq 1$, we let

$$\Delta \triangleq \nabla\hat{F}^i(p) - \nabla\hat{F}^i(p'),$$

where $p' \in \tilde{G}_s^{l-1}$ is the parent of $p$. Note that $\Delta$ is a $d$-dimensional vector whose entries are in the range $\big(2^{-l}\sqrt{d}\log(mn)/\sqrt{n}\big) \times \big[-1, +1\big]$. This is due to the Lipschitz continuity of the derivative of the functions in $\mathcal{F}$ (cf. Assumption 1) and the fact that $\|p - p'\| = 2^{-l}\sqrt{d}\log(mn)/\sqrt{n}$. Hence, we can use $O(d\log(mn))$ bits to represent $\Delta$ within accuracy $2\delta\log(mn)/\sqrt{n}$.

At the server, we choose an $s^* \in G$ that has the largest number of occurrences in the received signals. Then, base on the signals corresponding to $\tilde{G}_{s^*}^0$, we approximate the gradient of $F$ at $s^*$ as

$$\hat{\nabla}F(s^*) = \frac{1}{N_{s^*}} \sum_{\substack{\text{Signals of the form} \\ Y^i = (s^*, s^*, \Delta)}} \Delta,$$

where $N_{s^*}$ is the number of signals containing $s^*$ in the part $p$. Then, for any point $p \in \tilde{G}^l_{s^*}$ with $l \geq 1$, we compute

$$\hat{\nabla}F(p) = \hat{\nabla}F(p') + \frac{1}{N_p} \sum_{\substack{\text{Signals of the form} \\ Y^i = (s^*, p, \Delta)}} \Delta, \qquad (6)$$

where $N_p$ is the number of signals having point $p$ in their second argument. Finally, the sever lets $\hat{\theta}$ be a grid point $p$ in $\tilde{G}^t_{s^*}$ with the smallest $\|\hat{\nabla}F(p)\|$.

In the MRE-C-log algorithm the signals are of length $d/(d+1)\log m + d\log n$ bits, which is no larger than $d\log mn$. Please refer to Section 5 for discussions on how the MRE-C-log algorithm can be extended to work under more general communication constraints.

**Theorem 1** *Let $\hat{\theta}$ be the output of the above algorithm. Then,*

$$\Pr\left(\|\hat{\theta} - \theta^*\| > \frac{8d \, \log^{\frac{5}{\max(d,2)}+1}(mn)}{\lambda \, m^{\frac{1}{\max(d,2)}} n^{\frac{1}{2}}}\right) = \exp\left(-\Omega\big(\log^2(mn)\big)\right).$$

The proof is given in Appendix C. The proof goes by first showing that $s^*$ is a closest grid point of $G$ to $\theta^*$ with high probability. We then show that for any $l \leq t$ and any $p \in \tilde{G}^l_{s^*}$, the number of received signals corresponding to $p$ is large enough so that the server obtains a good approximation of $\nabla F$ at $p$. Once we have a good approximation $\hat{\nabla}F$ of $\nabla F$ at all points of $\tilde{G}^t_{s^*}$, a point at which $\hat{\nabla}F$ has the minimum norm lies close to the minimizer of $F$.

**Corollary 1** *Let $\hat{\theta}$ be the output of the above algorithm. There is a constant $\eta > 0$ such that for any $k \in \mathbb{N}$,*

$$\mathbb{E}\big[\|\hat{\theta} - \theta^*\|^k\big] < \eta \left(\frac{8d \, \log^{\frac{5}{\max(d,2)}+1}(mn)}{\lambda \, m^{\frac{1}{\max(d,2)}} n^{\frac{1}{2}}}\right)^k.$$

*Moreover, $\eta$ can be chosen arbitrarily close to 1, for large enough values of $mn$.*

The upper bound in Theorem 1 matches the lower bound in Theorem 2 of [Salehkaleybar et al., 2019] up to a polylogarithmic factor. In this view, the MRE-C-log algorithm has order optimal error. Moreover, as we show in Appendix C, in the course of computations, the server obtains an approximation $\hat{F}$ of $F$ such that for any $\theta$ in the cube $C_{s^*}$, we have $\|\nabla\hat{F}(\theta) - \nabla F(\theta)\| = \tilde{O}\big(m^{-1/d}n^{-1/2}\big)$. Therefore, the server not only finds the minimizer of $F$, but also obtains an approximation of $F$ at all points inside $C_{s^*}$. In the special case that $n = 1$, we have $C_{s^*} = [-1, 1]^d$, and as a result, the server would acquire an approximation of $F$ over the entire domain. This observation suggests the following insight: In the extreme distributed case ($n = 1$), finding an $O\big(m^{-1/d}\big)$-accurate minimizer of $\nabla F$ is as hard as finding an $O\big(m^{-1/d}\big)$-accurate approximation of $F$ for all points in the domain.

## 4   Experiments

We evaluated the performance of MRE-C-log on two learning tasks and compared with the averaging method (AVGM) in [Zhang et al., 2012]. Recall that in AVGM, each machine sends the empirical risk minimizer of its own data to the server and the average of received parameters at the server is returned in the output.

The first experiment concerns the problem of ridge regression. Here, each sample $(X, Y)$ is generated based on a linear model $Y = X^T\theta^* + E$, where $X$, $E$, and $\theta^*$ are sampled from $N(\mathbf{0}, I_{d\times d})$, $N(0, 0.01)$, and uniform distribution over $[0, 1]^d$, respectively. We consider square loss function with $l_2$ norm regularization: $f(\theta) = (\theta^T X - Y)^2 + 0.1\|\theta\|_2^2$. In the second experiment, we perform a logistic regression task, considering sample vector $X$ generated according to $N(\mathbf{0}, I_{d\times d})$ and labels $Y$ randomly drawn from $\{-1, 1\}$ with probability $\Pr(Y = 1|X, \theta^*) = 1/(1 + \exp(-X^T\theta^*))$. In both experiments, we consider a two dimensional domain ($d = 2$) and assumed that each machine has access to one sample ($n = 1$).

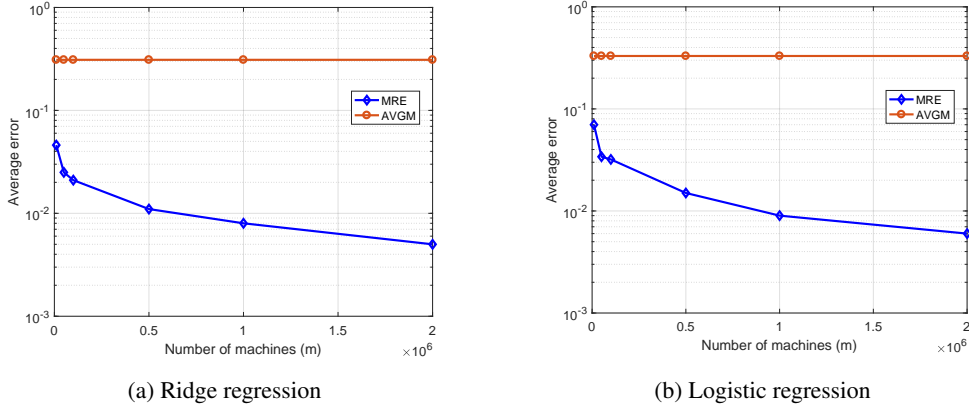

|  |  |
|:---:|:---:|
| (a) Ridge regression | (b) Logistic regression |

Figure 3: The average of MRE-C-log and AVGM algorithms versus the number of machines in two different learning tasks.

In Fig. 3, the average of $\|\hat{\theta} - \theta^*\|_2$ is computed over 100 instances for the different number of machines in the range $[10^4, 10^6]$. Both experiments suggest that the average error of MRE-C-log keep decreasing as the number of machines increases. This is consistent with the result in Theorem 1, according to which the expected error of MRE-C-log is upper bounded by $\tilde{O}(1/\sqrt{mn})$. It is evident from the error curves that MRE-C-log outperforms the AVGM algorithm in both tasks. This is because where $m$ is much larger than $n$, the expected error of the AVGM algorithm typically scales as $O(1/n)$, independent of $m$.

## 5 Discussion

We studied the problem of statistical optimization in a distributed system with one-shot communications. We proposed an algorithm, called MRE-C-log , with $O\big(\log(mn)\big)$-bits per message, and showed that its expected error is optimal up to a poly-logarithmic factor. Aside from being order optimal, the MRE-C-log algorithm has the advantage over the existing estimators that its error tends to zero as the number of machines goes to infinity, even when the number of samples per machine is upper bounded by a constant. This property is in line with the out-performance of the MRE-C-log algorithm in the $m \gg n$ regime, as discussed in our experimental results.

The main idea behind the MRE-C-log algorithm is that it essentially computes, in an efficient way, an approximation of the gradient of the expected loss over the entire domain. It then outputs a norm-minimizer of this approximate gradients, as an estimate of the minimizer of the expected loss. Therefore, MRE-C-log carries out the intricate and seemingly redundant task of approximating the loss function for all points in the domain, in order to resolve the apparently much easier problem of finding a single approximate minimizer for the loss function. In this view, it is quite counter-intuitive that such algorithm is order optimal in terms of expected error and sample complexity. This observation provides the interesting insight that in a distributed system with one shot communication, finding an approximate minimizer is as hard as finding an approximation of the function derivatives for all points in the domain.

Our algorithms and bounds are designed and derived for a broader class of functions with Lipschitz continuous first order derivatives, compared to the previous works that consider function classes with Lipschitz continuous second or third order derivatives. The assumption is indeed both practically important and technically challenging. For example, it is well-known that the loss landscapes involved in learning applications and neural networks are highly non-smooth. Therefore, relaxing assumptions on higher order derivatives is actually a practically important improvement over the previous works. On the other hand, considering Lipschitzness only for the first order derivative renders the problem way more difficult. To see this, note that when $n > m$, the existing upper bound $O(1/\sqrt{mn} + 1/n)$ for the case of Lipschitz second derivatives goes below the $O(m^{1/d}n^{1/2})$ lower bound in the case of Lipschitz first derivatives.

A drawback of the MRE-C-$\log$ algorithm is that each machine requires to know $m$ in order to set the number of levels for the grids. This however can be resolved by considering infinite number of levels, and letting the probability that $p$ is chosen from level $l$ decrease exponentially with $l$. Moreover, although communication budget of the MRE-C-$\log$ algorithm is $O(d \log mn)$ bits per signal, the algorithm can be extended to work under more general communication constraints, via dividing each signal to subsignals of length $O(d \log mn)$ each containing an independent independent signal of the MRE-C-$\log$ algorithm. The expected loss of this modified algorithm can be shown to still matches the existing lower bounds up to logarithmic factors. Please refer to Salehkaleybar et al. [2019] for a thorough treatment.

We also proposed, for $d = 1$, an algorithm with communication budget of one bit per transmission, whose error tends to zero in a rate of $O\left(1/\sqrt{m} + 1/\sqrt{n}\right)$ as $m$ and $n$ go to infinity simultaneously. We conjecture that this algorithms is order-optimal, in the sense that no randomized constant-bit algorithm has expected error smaller than $O\left(1/\sqrt{m} + 1/\sqrt{n}\right)$.

There are several open problems and directions for future research. The first group of problems involve the constant bit regime. It would be interesting if one could verify whether or not the bound in Proposition 1 is order optimal. Moreover, the constant bit algorithm in Subsection 3.1 is designed for one-dimensional domains and one-bit per transmission. Decent extensions of this algorithm to higher dimensions with vanishing errors under one bit per transmission constraint seem to be non-trivial. Investigating the power of more bits per transmission (constants larger than one bit) in reducing the expected error is another interesting direction.

Another important group of problems concerns the more restricted class of functions with Lipschitz continuous second order derivatives. Despite several attempts in the literature, the optimal scaling of expected error for this class of functions in the $m \gg n$ regime is still an open problem.

## Acknowledgments

This research was supported by Iran National Science Foundation (INSF) under contract No. 97012846.

## Footnotes

[1]Consider two convex functions $f_0(\theta) = \theta^2 + \theta^3/6$ and $f_1(\theta) = (\theta-1)^2 + (\theta-1)^3/6$ over $[0, 1]$. Consider a distribution $P$ that associates probability $1/2$ to each function. Then, $\mathbb{E}_P[f(\theta)] = f_0(\theta)/2 + f_1(\theta)/2$, and the optimal solution is $\theta^* = (\sqrt{15} - 3)/2 \approx 0.436$. On the other hand, in the averaging method proposed in [Zhang et al., 2012], assuming $n = 1$, the empirical minimizer of each machine is either $0$ if it observes $f_0$, or $1$ if it observes $f_1$. Therefore, the server receives messages $0$ and $1$ with equal probability , and $\mathbb{E}\big[\hat{\theta}\big] = 1/2$. Hence, $\mathbb{E}\big[|\hat{\theta} - \theta^*|\big] > 0.06$, for all values of $m$.

[2]The considered model here is similar to the one in [Salehkaleybar et al., 2019].

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
