[Supplementary Material]

# Appendices

## A Proof of Proposition 1

For simplicity, in this proof we will be working with the $[0, 1]$ interval as the domain. Consider the following randomized algorithm:

- Suppose that each machine $i$ observes $n$ function $f_1^i, \cdots, f_n^i$ and finds the minimizer of $\sum_{j=1}^n f_j^i(\theta)$, which we denote by $\theta^i$. Machine $i$ then sends a signal $Y^i$ of the following form

$$Y^i = \begin{cases} 0, & \text{with prob. } \theta^i, \\ 1, & \text{with prob. } 1 - \theta^i. \end{cases}$$

- The server receives signals from all machines and outputs $\hat{\theta} = 1/m \sum_{i=1}^m Y^i$.

For the above algorithm, we have

$$\text{var}\left(\hat{\theta}\right) = \text{var}\left(\frac{1}{m}\sum_{i=1}^m Y^i\right) = \frac{1}{m}\text{var}\left(Y^1\right) = O\left(\frac{1}{m}\right), \tag{7}$$

where the last equality is because $Y^1$ is a binary random variable. Then,

$$\begin{aligned}
\mathbb{E}\left[\left(\hat{\theta} - \theta^*\right)^2\right] &= \mathbb{E}\left[\left(\hat{\theta} - \mathbb{E}[\hat{\theta}] + \mathbb{E}[\hat{\theta}] - \theta^*\right)^2\right] \\
&= \mathbb{E}\left[\left(\hat{\theta} - \mathbb{E}[\hat{\theta}]\right)^2\right] + \mathbb{E}\left[\left(\mathbb{E}[\hat{\theta}] - \theta^*\right)^2\right] \\
&= \text{var}\left(\hat{\theta}\right) + \left(\mathbb{E}[\hat{\theta} - \theta^*]\right)^2 \\
&= O\left(\frac{1}{m}\right) + O\left(\frac{1}{n}\right),
\end{aligned}$$

where the last equality is due to (7) and Lemma 1. This completes the proof of Proposition 1.

## B Proof of Proposition 2

For the ease of notation, we denote the number of grid points by $k = \sqrt[3]{m}/\log(m)$. Then, the probability of choosing each point $\theta$ in the grid equals $1/k$. Let $N(\theta)$ be the number of machines that select grid point $\theta$. From Hoeffding's inequality (cf. Lemma 2 (a)), for any grid point $\theta$,

$$\begin{aligned}
\Pr\left(N(\theta) < \frac{m}{2k}\right) &= \Pr\left(-N(\theta) + \frac{m}{k} > \frac{m}{2k}\right) \\
&\leq \Pr\left(\frac{1}{m}\left|N(\theta) - \mathbb{E}[N(\theta)]\right| > \frac{1}{2k}\right) \\
&\leq \exp\left(-2m\left(\frac{1}{2k}\right)^2\right) \\
&= \exp\left(\frac{-\sqrt[3]{m}\log^2 m}{2}\right).
\end{aligned}$$

It then follows from union bound that with probability at most $1 - k\exp\left(-\Omega(\sqrt[3]{m})\right) = 1 - \exp\left(-\Omega(\sqrt[3]{m})\right)$, for any grid point $\theta$, we have $N(\theta) \geq m/(2k)$. For the rest of the proof, we assume that $N(\theta) \geq m/(2k)$, for all grid points $\theta$.

Recall that the derivatives of the functions in $\mathcal{F}$ are assumed to be in the $[-1, 1]$ interval (cf. Assumption 1). Then, it follows from the Hoeffding's inequality (cf. Lemma 2 (a)) that for any point $\theta$ in the grid and any $\alpha \geq 1$,

$$\Pr\left(\left|\hat{F}'(\theta) - F'(\theta)\right| > \frac{\alpha}{k}\right) \leq 2\exp\left(\frac{-2}{2^2} \times \frac{m}{2k} \times \left(\frac{\alpha}{k}\right)^2\right) = 2\exp\left(-\frac{\alpha^2\log^2 m}{4}\right).$$

Let $\mathcal{E}_\alpha$ be the event that for all grid points $\theta$, $\left|\hat{F}'(\theta) - F'(\theta)\right| \leq \alpha/k$. From union bound, we have

$$1 - \Pr(\mathcal{E}_\alpha) \leq 2k \exp\left(-\frac{\alpha^2 \log^3 m}{4}\right) = O\left(\exp\left(-\alpha^2 \log^3 m\right)\right), \tag{8}$$

where the equality is because $\alpha \geq 1$. Let $\tilde{\theta}$ be the closest grid point to $\theta^*$. Then, $|\tilde{\theta} - \theta^*| \leq 1/k$ and it follows from the Lipschitz continuity of $F'$ that

$$\left|F'(\tilde{\theta})\right| = \left|F'(\tilde{\theta}) - F'(\theta^*)\right| \leq |\tilde{\theta} - \theta^*| \leq \frac{1}{k}. \tag{9}$$

Since $F$ is $\lambda$-strongly convex, for any $\alpha \geq 1$, we have

$$\Pr\left(|\hat{\theta} - \theta^*| > \frac{3\alpha}{k\lambda}\right) \leq \Pr\left(\left|F'(\hat{\theta}) - F'(\theta^*)\right| > \frac{3\alpha}{k}\right)$$

$$= \Pr\left(\left|F'(\hat{\theta})\right| > \frac{3\alpha}{k}\right)$$

$$\leq \Pr\left(\left|F'(\hat{\theta}) - \hat{F}'(\hat{\theta})\right| > \frac{\alpha}{k}\right) + \Pr\left(\left|\hat{F}'(\hat{\theta})\right| > \frac{2\alpha}{k}\right)$$

$$\leq^a O\left(\exp\left(-\alpha^2 \log^3 m\right)\right) + \Pr\left(\left|\hat{F}'(\hat{\theta})\right| > \frac{2\alpha}{k}\right)$$

$$\leq^b O\left(\exp\left(-\alpha^2 \log^3 m\right)\right) + \Pr\left(\left|\hat{F}'(\tilde{\theta})\right| > \frac{2\alpha}{k}\right)$$

$$\leq O\left(\exp\left(-\alpha^2 \log^3 m\right)\right) + \Pr\left(\left|\hat{F}'(\tilde{\theta}) - F'(\tilde{\theta})\right| > \frac{\alpha}{k}\right) + \Pr\left(\left|F'(\tilde{\theta})\right| > \frac{\alpha}{k}\right)$$

$$\leq^c O\left(\exp\left(-\alpha^2 \log^3 m\right)\right) + O\left(\exp\left(-\alpha^2 \log^3 m\right)\right) + 0$$

$$= O\left(\exp\left(-\alpha^2 \log^3 m\right)\right),$$

where $(a)$ follows from (8), $(b)$ is because $\hat{\theta}$ minimizes $\left|\hat{F}'(\theta)\right|$ over all grid points $\theta$, and $(c)$ is due to (8) and (9). This completes the proof of Proposition 2.

## C  Proof of Theorem 1

We first show that $s^*$ is a closest grid point of $G$ to $\theta^*$ with high probability. We then show that for any $l \leq t$ and any $p \in \tilde{G}^l_{s^*}$, the number of received signals corresponding to $p$ is large enough so that the server obtains a good approximation of $\nabla F$ at $p$. Once we have a good approximation of $\nabla F$ at all points of $\tilde{G}^t_{s^*}$, a point with the minimum norm for this approximation lies close to the minimizer of $F$.

Suppose that machine $i$ observes functions $f_1^i, \ldots, f_n^i$. Recall the definition of $\theta^i$ in (3). The following lemma provides a bound on $\theta^i - \theta^*$, which improves upon the bound in Lemma 8 of [Zhang et al., 2013].

**Lemma 1** *For any $i \leq m$,*

$$\Pr\left(\|\theta^i - \theta^*\| \geq \frac{\alpha}{\sqrt{n}}\right) \leq d \exp\left(\frac{-\alpha^2 \lambda^2}{d}\right),$$

*where $\lambda$ is the lower bound on the curvature of $F$ (cf. Assumption 1).*

The proof relies on concentration inequalities, and is given in Appendix D. We collect two well-known concentration inequalities in the following lemma.

**Lemma 2** *(Concentration inequalities)*

(a) *(Hoeffding's inequality) Let $X_1, \cdots, X_n$ be independent random variables ranging over the interval $[a, a + \gamma]$. Let $\bar{X} = \sum_{i=1}^n X_i/n$ and $\mu = \mathbb{E}[\bar{X}]$. Then, for any $\alpha > 0$,*

$$\Pr\left(|\bar{X} - \mu| > \alpha\right) \leq 2 \exp\left(\frac{-2n\alpha^2}{\gamma^2}\right).$$

(b) *(Theorem 4.2 in Motwani and Raghavan [1995]) Let $X_1, \cdots, X_n$ be independent Bernoulli random variables, $X = \sum_{i=1}^{n} X_i$, and $\mu = \mathbb{E}[X]$. Then, for any $\alpha \in (0, 1]$,*

$$\Pr\left(X < (1-\alpha)\mu\right) \leq \exp\left(-\frac{\mu\alpha^2}{2}\right).$$

Based on the above lemma, we have

$$
\begin{aligned}
\Pr\left(\|\theta^i - \theta^*\| \leq \frac{\log(mn)}{2\sqrt{n}}, \text{ for } i = 1, \ldots, m\right) \\
\geq 1 - m \Pr\left(\|\theta^1 - \theta^*\| \geq \frac{\log(mn)}{2\sqrt{n}}\right) \\
\geq 1 - md \exp\left(\frac{-\lambda^2 \log^2(mn)}{4d}\right) \\
= 1 - \exp\left(-\Omega\left(\log^2(mn)\right)\right),
\end{aligned}
\tag{10}
$$

where the first inequality is due to the union bound and the fact that the distributions of $\theta^1, \ldots, \theta^m$ are identical, and the second inequality follows from Lemma 1. Thus, with probability at least $1 - \exp\left(-\Omega(\log^2(mn))\right)$, every $\theta^i$ is in the distance $\log(mn)/2\sqrt{n}$ from $\theta^*$. For each machine $i$, let $s^i$ be the $s$-component of machine $i$'s signal. Therefore, with probability at least $1 - \exp\left(-\Omega(\log^2(mn))\right)$, for any machine $i$,

$$
\begin{aligned}
\Pr\left(\|s^i - \theta^*\|_\infty > \frac{\log(mn)}{\sqrt{n}}\right) &\leq \Pr\left(\|s^i - \theta^i\|_\infty + \|\theta^i - \theta^*\|_\infty > \frac{\log(mn)}{\sqrt{n}}\right) \\
&\leq \Pr\left(\|s^i - \theta^i\|_\infty > \frac{\log(mn)}{2\sqrt{n}}\right) \\
&\quad + \Pr\left(\|\theta^i - \theta^*\|_\infty > \frac{\log(mn)}{2\sqrt{n}}\right) \\
&= 0 + \Pr\left(\|\theta^i - \theta^*\|_\infty > \frac{\log(mn)}{2\sqrt{n}}\right) \\
&= \exp\left(-\Omega\left(\log^2(mn)\right)\right),
\end{aligned}
$$

where the first equality is due to the choice of $s^i$ as the nearest grid point to $\theta^i$, and the last equality follows from (10). Recall that $s^*$ is the grid point with the largest number of occurrences in the received signals. Therefore, with probability at least $1 - \exp\left(-\Omega(\log^2(mn))\right)$,

$$\|s^* - \theta^*\|_\infty \leq \frac{\log(mn)}{\sqrt{n}};
\tag{11}$$

equivalently, $\theta^*$ lies in the $\left(2\log(mn)/\sqrt{n}\right)$-cube $C_{s^*}$ centered at $s^*$.

Let $m^*$ be the number of machines that select $s = s^*$. We let $\mathcal{E}'$ be the event that $m^* \geq m/2^d$. Since the grid $G$ has block size $2\log(mn)/\sqrt{n}$, there are at most $2^d$ points $s$ of the grid that satisfy $\|s - \theta^*\|_\infty \leq \log(mn)/\sqrt{n}$. It then follows from (11) that

$$\Pr\left(\mathcal{E}'\right) = 1 - \exp\left(-\Omega(\log^2(mn))\right).
\tag{12}$$

We now turn our focus to the inside of cube $C_{s^*}$. Let

$$\epsilon \triangleq \frac{2\sqrt{d}\log(mn)}{\sqrt{n}} \times \delta = \frac{8d \log^{1 + \frac{5}{\max(d,2)}}(mn)}{n^{\frac{1}{2}} m^{\frac{1}{\max(d,2)}}}.
\tag{13}$$

For any $p \in \bigcup_{l \leq t} \tilde{G}_{s^*}^l$, let $N_p$ be the number of machines that select point $p$. Let $\mathcal{E}''$ be the event that for any $l \leq t$ and any $p \in \tilde{G}_{s^*}^l$, we have

$$N_p \geq \frac{d^2 2^{-2l} \log^6(mn)}{2n\epsilon^2}.
\tag{14}$$

Then,

**Lemma 3** $\Pr\left(\mathcal{E}''\right) = 1 - \exp\left(-\Omega(\log^2(mn))\right).$

The proof is based on the concentration inequality in Lemma 2 (b), and is given in Appendix E.

Capitalizing on Lemma 3, we now obtain a bound on the estimation error of gradient of $F$ at the grid points in $\tilde{G}_{s*}^l$. Let $\mathcal{E}'''$ be the event that for any $l \leq t$ and any grid point $p \in \tilde{G}_{s*}^l$, we have

$$\left\|\hat{\nabla}F(p) - \nabla F(p)\right\| < \frac{\epsilon}{4}.$$

**Lemma 4** $\Pr\left(\mathcal{E}'''\right) = 1 - \exp\left(-\Omega(\log^2(mn))\right).$

The proof is given in Appendix F and relies on Hoeffding's inequality and the lower bound on the number of received signals for each grid point, driven in Lemma 3.

In the remainder of the proof, we assume that (11) and $\mathcal{E}'''$ hold. Let $p^*$ be the closest grid point in $\tilde{G}_{s*}^t$ to $\theta^*$. Therefore,

$$\|p^* - \theta^*\| \leq \sqrt{d}\,2^{-t}\frac{\log(mn)}{\sqrt{n}} = \epsilon/2. \tag{15}$$

Then, it follows from $\mathcal{E}'''$ that

$$\begin{aligned}
\|\hat{\nabla}F(p^*)\| &\leq \left\|\hat{\nabla}F(p^*) - \nabla F(p^*)\right\| + \|\nabla F(p^*)\| \\
&\leq \epsilon/4 + \|\nabla F(p^*)\| \\
&= \epsilon/4 + \left\|\nabla F(p^*) - \nabla F(\theta^*)\right\| \\
&\leq \epsilon/4 + \left\|p^* - \theta^*\right\| \\
&\leq \epsilon/4 + \epsilon/2 \\
&= 3\epsilon/4,
\end{aligned} \tag{16}$$

where the second inequality is due to $\mathcal{E}'''$, the third inequality follows from the Lipschitz continuity of $\nabla F$, and the last inequality is from (15). Therefore,

$$\begin{aligned}
\|\hat{\theta} - \theta^*\| &\leq \frac{1}{\lambda}\left\|\nabla F(\hat{\theta}) - \nabla F(\theta^*)\right\| \\
&= \frac{1}{\lambda}\|\nabla F(\hat{\theta})\| \\
&\leq \frac{1}{\lambda}\|\hat{\nabla}F(\hat{\theta})\| + \frac{1}{\lambda}\left\|\hat{\nabla}F(\hat{\theta}) - \nabla F(\hat{\theta})\right\| \\
&\leq^a \frac{1}{\lambda}\|\hat{\nabla}F(\hat{\theta})\| + \frac{\epsilon}{4\lambda} \\
&\leq^b \frac{1}{\lambda}\|\hat{\nabla}F(p^*)\| + \frac{\epsilon}{4\lambda} \\
&\leq^c \frac{3\epsilon}{4\lambda} + \frac{\epsilon}{4\lambda} \\
&= \frac{\epsilon}{\lambda},
\end{aligned}$$

(a) Due to event $\mathcal{E}'''$.
(b) Because the output of the server, $\hat{\theta}$, is a grid point $p$ in $\tilde{G}_{s*}^t$ with smallest $\|\hat{\nabla}F(p)\|$.
(c) According to (16).
Finally, it follows from (11) and Lemma 4 that the above inequality holds with probability $1 - \exp\left(-\Omega(\log^2(mn))\right)$. Equivalently,

$$\Pr\left(\|\hat{\theta} - \theta^*\| \geq \frac{\epsilon}{\lambda}\right) = \exp\left(-\Omega\left(\log^2(mn)\right)\right),$$

and Theorem 1 follows.

# D   Proof of Lemma 1

Let $F^i(\theta) = \sum_{j=1}^{n/2} f_j^i(\theta)$, for all $\theta \in [-1,1]^d$. From the lower bound $\lambda$ on the second derivative of $F$, we have

$$\|\nabla F(\theta^i) - \nabla F^i(\theta^i)\| \; = \; \|\nabla F(\theta^i)\| \; = \; \|\nabla F(\theta^i) - \nabla F(\theta^*)\| \; \geq \; \lambda\|\theta^i - \theta^*\|,$$

where the two equalities are because $\theta^i$ and $\theta^*$ are the the minimizers of $F^i$ and $F$, respectively. Then,

$$
\begin{aligned}
\Pr\left(\|\theta^i - \theta^*\| \geq \frac{\alpha}{\sqrt{n}}\right) \; &\leq \; \Pr\left(\|\nabla F(\theta^i) - \nabla F^i(\theta^i)\| \geq \frac{\lambda\alpha}{\sqrt{n}}\right) \\
&\leq^a \; \sum_{j=1}^{d} \Pr\left(\left|\frac{\partial F^i(\theta^i)}{\partial \theta_j} - \frac{\partial F(\theta^i)}{\partial \theta_j}\right| > \frac{\alpha\lambda}{\sqrt{d}\sqrt{n}}\right) \\
&= \; d\,\Pr\left(\left|\frac{2}{n}\sum_{l=1}^{n/2}\frac{\partial}{\partial\theta_j}f_l^i(\theta^i) - \mathbb{E}_{f\sim P}\left[\frac{\partial}{\partial\theta_j}f(\theta^i)\right]\right| \geq \frac{\alpha\lambda}{\sqrt{d}\sqrt{n}}\right) \\
&=^b \; d\exp\left(-\frac{\alpha^2\lambda^2}{d}\right),
\end{aligned}
$$

(17)

($a$) Follows from the union bound and the fact that for any $d$-dimensional vector $v$, there exists an entry $v_i$ such that $\|v\| \leq |v_i|/\sqrt{d}$.
($b$) Due to Hoeffding's inequality (cf. Lemma 2 (a)).
This completes the proof of Lemma 1.

# E   Proof of Lemma 3

Recall that for any $l \leq t$, given $s = s^*$, the probability that $p \in \tilde{G}_{s^*}^l$ is $2^{(d-2)l}/\sum_{j=0}^{t}2^{(d-2)j}$. Assuming $\mathcal{E}'$, for any $l \leq t$ and any $p \in \tilde{G}_{s^*}^l$,

$$
\begin{aligned}
\mathbb{E}[N_p] \; &= \; 2^{-dl} \times \frac{2^{(d-2)l}}{\sum_{j=0}^{t}2^{(d-2)j}} \times m^* \\
&= \; \frac{2^{-2l}m^*}{\sum_{j=0}^{t}2^{(d-2)j}} \\
&\geq^a \; \frac{2^{-2l}m}{2^d} \times \frac{1}{\sum_{j=0}^{t}2^{(d-2)j}} \\
&\geq^b \; \frac{2^{-2l}m}{2^d} \times \frac{1}{t2^{t\max(0,d-2)}} \\
&=^c \; \frac{2^{-2l}m}{2^d} \times \frac{\delta^{\max(0,d-2)}}{\log(1/\delta)} \\
&\geq^d \; \frac{2^{-2l}m}{2^d\delta^2} \times \frac{\delta^{\max(d,2)}}{\log(mn)} \\
&=^e \; \frac{2^{-2l}m}{2^d\delta^2\log(mn)} \times (4\sqrt{d})^{\max(d,2)}\frac{\log^5(mn)}{m} \\
&\geq \; \frac{4d2^{-2l}\log^4(mn)}{\delta^2},
\end{aligned}
$$

(18)

where ($a$) follows from $\mathcal{E}'$, ($b$) is valid for all non-negative integers $t$ and $d$, ($c$) is from the definition of $t = \log(1/\delta)$, ($d$) is due to the fact that $1/\delta \leq \sqrt{m} \leq mn$ and ($e$) is because of the definition of $\delta$

in (4). Then,

$$\Pr\left(N_p \le \frac{d^2 2^{-2l}\log^6(mn)}{2n\epsilon^2}\right) = \Pr\left(N_p \le \frac{1}{2}\times\frac{d2^{-2l}\log^4(mn)}{4\delta^2}\right)$$

$$\le \Pr\left(N_p \le \frac{\mathbb{E}[N_p]}{2}\right) \tag{19}$$

$$\le 2^{-(1/2)^2 \mathbb{E}[N_p]/2}$$

$$= \exp\big(-\Omega(\log^4(mn))\big),$$

where the first equality is from the definition of $\epsilon$ in (13), the first inequality is due to (18), the second inequality follows from Lemma 2 (b), and the last equality is due to (18) and the fact that $2^{-2l_p} \ge 2^{-2t} = 2^{-2\log(1/\delta)} = \delta^2$. Then,

$$\Pr\left(\mathcal{E}'' \mid \mathcal{E}'\right) \ge 1 - \sum_{l=0}^{t}\sum_{p\in\tilde{G}_{s*}^l}\Pr\left(N_p \le \frac{d^2 2^{-2l_p}\log^6(mn)}{2n\epsilon^2}\right)$$

$$\ge 1 - t2^{dt}\exp\big(-\Omega\big(\log^4(mn)\big)\big)$$

$$\ge 1 - \log(1/\delta)\left(\frac{1}{\delta}\right)^d\exp\big(-\Omega\big(\log^4(mn)\big)\big)$$

$$> 1 - m\log(m)\exp\big(-\Omega\big(\log^4(mn)\big)\big)$$

$$= 1 - \exp\big(-\Omega\big(\log^4(mn)\big)\big).$$

On the other hand, we have from (12) that $\Pr\left(\mathcal{E}'\right) = 1 - \exp\big(-\Omega(\log^2(mn))\big)$. Then, $\Pr\left(\mathcal{E}''\right) = 1 - \exp\big(-\Omega(\log^2(mn))\big)$ and Lemma 3 follows.

## F   Proof of Lemma 4

For any $l \le t$ and any $p \in \tilde{G}_{s*}^0$, let

$$\hat{\Delta}(p) = \frac{1}{N_p}\sum_{\substack{\text{Signals of the form}\\ Y^i=(s^*,p,\Delta)}}\Delta,$$

and let $\Delta^*(p) = \mathbb{E}[\hat{\Delta}(p)]$.

We first consider the case $l = 0$. Note that $\tilde{G}_{s*}^0$ consists of a single point $p = s^*$. Moreover, the component $\Delta$ in each signal is the average over the gradient of $n/2$ independent functions. Then, $\hat{\Delta}(p)$ is the average over the gradient of $N_p \times n/2$ independent functions. Given event $\mathcal{E}''$, for any entry $j$ of the gradient, it follows from Hoeffding's inequality (Lemma 2 (a)) that

$$\Pr\left(\left|\hat{\Delta}_j(p) - \Delta_j^*(p)\right| \ge \frac{\epsilon}{4\sqrt{d}\log(mn)}\right)$$

$$\le \exp\left(-N_p n\times\left(\frac{\epsilon}{4\sqrt{d}\log(mn)}\right)^2/2^2\right)$$

$$\le \exp\left(-n\frac{d^2\log^6(mn)}{8n\epsilon^2}\times\frac{\epsilon^2}{16d\log^2(mn)}\right) \tag{20}$$

$$= \exp\left(\frac{-d\log^4(mn)}{128}\right)$$

$$= \exp\big(-\Omega\big(\log^4(mn)\big)\big).$$

For $l \ge 1$, consider a grid point $p \in \tilde{G}_{s*}^l$ and let $p'$ be the parent of $p$. Then, $\|p - p'\| = \sqrt{d}\,2^{-l}\log(mn)/\sqrt{n}$. Furthermore, for any function $f \in \mathcal{F}$, we have $\|\nabla f(p) - \nabla f(p')\| \le \|p - p'\|$.

Hence, for any $j \leq n$,

$$\left| \frac{\partial f(p)}{\partial x_j} - \frac{\partial f(p')}{\partial x_j} \right| \leq \|p - p'\| = \frac{\sqrt{d} \log(mn) 2^{-l}}{\sqrt{n}}.$$

Therefore, $\hat{\Delta}_j(p)$ is the average of $N_p \times n/2$ independent variables with absolute values no larger than $\gamma \triangleq \sqrt{d} \log(mn) 2^{-l}/\sqrt{n}$. Given event $\mathcal{E}''$, it then follows from the Hoeffding's inequality that

$$\Pr\left( \left| \hat{\Delta}_j(p) - \Delta_j^*(p) \right| \geq \frac{\epsilon}{4\sqrt{d}\log(mn)} \right)$$

$$\leq \exp\left( -nN_p \times \frac{1}{(2\gamma)^2} \times \left( \frac{\epsilon}{4\sqrt{d}\log(mn)} \right)^2 \right)$$

$$\leq^a \exp\left( -n \frac{d^2 2^{-2l} \log^6(mn)}{2n\epsilon^2} \times \frac{n}{4d 2^{-2l} \log^2(mn)} \times \frac{\epsilon^2}{16d \log^2(mn)} \right)$$

$$= \exp\left( -n\log^2(mn)/128 \right)$$

$$= \exp\left( -\Omega\left( \log^2(mn) \right) \right),$$

where the second inequality is by substituting $N_p$ from (14). Employing union bound, we obtain

$$\Pr\left( \left\| \hat{\Delta}(p) - \Delta^*(p) \right\| \geq \frac{\epsilon}{4\log(mn)} \right)$$

$$\leq \sum_{j=1}^{d} \Pr\left( \left| \hat{\Delta}_j(p) - \Delta_j^*(p) \right| \geq \frac{\epsilon}{4\sqrt{d}\log(mn)} \right)$$

$$= d \exp\left( -\Omega\left( \log^2(mn) \right) \right)$$

$$= \exp\left( -\Omega\left( \log^2(mn) \right) \right).$$

Recall from (6) that for any non-zero $l \leq t$ and any $p \in \tilde{G}_{s^*}^l$ with parent $p'$,

$$\hat{\nabla} F(p) - \nabla F(p) = \hat{\nabla} F(p') - \nabla F(p') + \hat{\Delta}(p) - \Delta^*(p).$$

Then,

$$\Pr\left( \left\| \hat{\nabla} F(p) - \nabla F(p) \right\| > \frac{(l+1)\epsilon}{4\log(mn)} \right)$$

$$\leq \Pr\left( \left\| \hat{\nabla} F(p') - \nabla F(p') \right\| > \frac{l\epsilon}{4\log(mn)} \right)$$

$$+ \Pr\left( \left\| \hat{\Delta}(p) - \Delta^*(p) \right\| > \frac{\epsilon}{4\log(mn)} \right)$$

$$\leq \Pr\left( \left\| \hat{\nabla} F(p') - \nabla F(p') \right\| > \frac{l\epsilon}{4\log(mn)} \right) + \exp\left( -\Omega\left( \log^2(mn) \right) \right).$$

Employing an induction on $l$, we obtain for any $l \leq t$,

$$\Pr\left( \left\| \hat{\nabla} F(p) - \nabla F(p) \right\| > \frac{(l+1)\epsilon}{4\log(mn)} \right) \leq \exp\left( -\Omega\left( \log^2(mn) \right) \right).$$

Therefore, for any grid point $p$,

$$\Pr\left( \left\| \hat{\nabla} F(p) - \nabla F(p) \right\| > \frac{\epsilon}{4} \right) \leq \Pr\left( \left\| \hat{\nabla} F(p) - \nabla F(p) \right\| > \frac{(t+1)\epsilon}{4\log(mn)} \right)$$

$$= \exp\left( -\Omega\left( \log^2(mn) \right) \right),$$

where the inequality is because $t + 1 = log(1/\delta) + 1 \leq \log(mn)$. It then follows from the union bound that

$$\Pr\left(\mathcal{E}''' \mid \mathcal{E}''\right) \geq 1 - \sum_{l=0}^{t} \sum_{p \in \tilde{G}_{s*}^l} \Pr\left(\left\|\hat{\nabla}F(p) - \nabla F(p)\right\| > \frac{\epsilon}{4}\right)$$

$$\geq 1 - t2^{dt} \exp\left(-\Omega\left(\log^2(mn)\right)\right)$$

$$= 1 - \log(1/\delta)\left(\frac{1}{\delta}\right)^d \exp\left(-\Omega\left(\log^2(mn)\right)\right)$$

$$\geq 1 - m\log(m)\exp\left(-\Omega\left(\log^2(mn)\right)\right)$$

$$= 1 - \exp\left(-\Omega\left(\log^2(mn)\right)\right).$$

On the other hand, we have from Lemma 3 that $\Pr\left(\mathcal{E}''\right) = 1 - \exp\left(-\Omega(\log^2(mn))\right)$. Then, $\Pr\left(\mathcal{E}'''\right) = 1 - \exp\left(-\Omega(\log^2(mn))\right)$ and Lemma 4 follows.