[Reviews · NeurIPS 2019]

Reviewer 1



Originality: to the best of my knowledge, the proposed algorithm is new in this area. Unlike prior one-shot distributed learning algorithms whose statistical error rate does not approach 0 as m approaches infinity, the error rate of this algorithm decays with m. The decay rate as a function of m is not very fast when d is large, (m^-1/d), but this matches the lower bound so there is not much we can improve. The design of the algorithm also has some interesting ideas on how to encode the local gradient information of the loss functions and communicate them using a limited number of bits. Quality: I think the paper is technically sound. The assumptions and main results are stated clearly. One point that I am not very satisfied is that it seems that there is a dimension dependence and it seems this dependence is hidden in the big-O notation. How does the final accuracy (or communication complexity) depend on d? Is it optimal or can we improve it? How do we compare it with prior works? Clarity: I think the paper is overally clearly written. It is easy to follow and I think I understand most part of it. Some minor comments to improve the clarity: Prior to Section 3.1, it seems that the function is defined in the space [-1, 1]^d. But in Section 3.1, it changed to [0, 1] without an explicit statement. Page 5, line 178, \theta_i -> \theta^i. Significance: I think there is some significant contribution in this paper. In addition to my prior comments, it would be great to improve the experiment section. Adding real-world datasets could be helpful. ========================= After author feedback: The authors clarified my concerns on the dimension dependence. I still hope to see some experiments on real-world datasets. I changed my score to 7.

Reviewer 2



The authors study the problem of statistical optimization in communication restricted setting where samples are spread among multiple machines and a central entity needs to compute an optimal estimator. Specifically, they study the one-shot model. They provide an asymptotically essentially optimal algorithm to solve the problem. They also adduce empirical results. The main idea in their algorithm is to use approximate gradients as a proxy for the actual expected loss. Other than that, the algorithm uses discretization tricks to compute a coarse approximation of all the loss of all functions in order to find the best estimate. The paper is well written in terms of clarity. The problem is certainly of importance, and the solution represents an important milestone in the search for distributed algorithms for the problem because it is nearly optimal. The proof of the communication upper bound requires careful work. The empirical evidence gives a final touch in convincing the veracity of the algorithm. While the general exposition is clear, the discussion of related work needs to be more thorough. The importance of the problem should be motivated more clearly. Also, the paper has a technical improvement wherein they relax the class of functions that are being studied. The authors need to elaborate on the importance of this assumption. Without that, a reader could construe it to be nothing more than a mild technical advantage. In conclusion, the problem is important, and the contribution is strong because the algorithm is near optimal. The analysis does not particularly involve new ideas, but it is non-trivial and requires care.

Reviewer 3



The paper considers the problem of convex optimization in the distributed setting where each of the m users observes n i.i.d samples from an unknown distribution and are allowed to transmit a one-shot message to the server. The server then tries to find the minimizer of the expectation of some convex loss function over certain parameter space. This problem has gathered a lot of interest recently, especially in the setting of federated learning, where the data are located on users' devices. In the setting where m >> n and each user can only send a limited number of \log(mn) bits, previous results based on sending local minimizers fail to achieve an error that goes to zero when m tends to infinity and n is fixed. The paper proposes multi-resolution estimator (MRE), which can achieve an error of \tilde{O} (m^{-1/d}n^{-1/2}) when d > 2 assuming the loss is bounded, smooth and strongly convex. This matches the lower bound in this setting up to logarithmic factors. The algorithm is based on the observation that when n is small, local minimizers don't carry all the information needed to compute the overall minimizer. The algorithm complements this with estimations of gradients on randomly selected points around the local minimizer. The final output is the point with the smallest gradient norm. The author also empirically demonstrates the result by two simple examples of ridge regression and logistic regression. The paper is well written with a clear statement of the result and analysis. I enjoy reading the paper. The result is cute with interesting observations. The result could have been stronger if the author can analyze the whole tradeoff between the communication budget and expected loss. Overall I would recommend the paper for acceptance. Minor comments: 1. There is are missing ellipsis in Fig 1. 2. Corollary 1 seems to be stated in the wrong direction. 3. When stating the communication budget, it would be nice to include d, the dimensionality in it or state the dependence clearly at the beginning. ==================== After feedback: Thank the authors to address my concern on the tradeoff when larger amount of communication budget is provided. I have increased my score from 6 to 7.

[Author Response · NeurIPS 2019]

We thank the reviewers for their valuable comments. We believe that their feedback, especially the comments on
communication budget, significantly improved the integration and presentation of the paper.

## Response to Reviewer 1.

**There is a dimension dependence and it seems this dependence is hidden in the big-$O$ notation. How does the**
**final accuracy (or communication complexity) depend on $d$? Is it optimal or can we improve it? How do we**
**compare it with prior works?** – The exact length of signals in the MRE algorithm is $d/(d+1)\log m + d\log n$ which
is no larger than $d\log mn$. Dependence of the accuracy on $d$ is already reflected in Theorem 1 and Corollary 1. Prior
works in [Zhang et al. 2012] and [Jordan et al. 2018], although not specifically mentioned, require communication
budgets of $O(d\log mn)$ and $O(dm)$ bits, respectively. Regarding optimality of the budget with respect to $d$, we are
not aware of a lower bound that reflects an explicit dependence on $d$. It is a very intriguing question whether or not
vanishing error is possible under signals of lengths sub-linear in $d$.

**Minor comments: a) Prior to Section 3.1, it seems that the function is defined in the space $[-1,1]^d$. But in**
**Section 3.1, it changed to $[0,1]$ without an explicit statement; b) Page 5, line 178, $\theta_i \rightarrow \theta^i$.** – a) Explicit statement
will be added to the finial submission. b) Corrected.

## Response to Reviewer 2.

**The paper has a technical improvement wherein they relax the class of functions that are being studied. It**
**is unclear that allowing functions that are Lipschitz continuous with continuous first order derivatives is of**
**terrific importance. The authors need to elaborate on the importance of this assumption.** – The assumption is
indeed both practically important and technically challenging. It is well-known that the loss landscapes involved in
learning applications and neural networks are highly non-smooth. For example, the loss surface of a neural network
with ReLU activations is non-differentiable. Even when the activation functions are differentiable, the loss surface of a
deep network would be far from smoothness. Therefore, relaxing assumptions on higher order derivatives is actually a
practically important improvement over the previous works.
On the other hand, the assumption brings in serious technical challenges. To see this, note that when $n > m$, the
existing upper bound $O(\sqrt{mn} + 1/n)$ for the case of Lipschitz second derivatives goes below the $O(m^{1/d}n^{1/2})$ lower
bound in the case of Lipschitz first derivatives. This shows that the first order derivative assumption makes the problem
way more difficult. We will add discussions on these points in the final submission.

**While the general exposition is clear, the discussion of related work needs to be more thorough. The importance**
**of the problem should be motivated more clearly.** – "The problem has gathered a lot of interest recently, especially
in the setting of federated learning, where the data are located on users' devices". We will make the motivation more
clear in the final submission.

**Study the problem in some other, more general, communication models.** – We think this is a very important
comment and take it seriously. Please refer to the response to the first question of Reviewer 3.

## Response to Reviewer 3.

**The result could have been stronger if the author can analyze the whole tradeoff between the communication**
**budget and expected loss.** – Thanks to this comment, we realized that by a simple modification, our algorithm can
handle the case of general communication budget. Better yet, the expected loss of this modified algorithm still matches
the existing lower bounds up to logarithmic factors. Below we briefly discuss this modification and the optimality of its
expected loss. We will revise the final version of the paper to reflect these facts.
The modified algorithm: Let $b$ be the communication budget (the number of bits) per signal. Each machine divides its
$b$-length signal into $b/(d\log mn)$ number of $(d\log mn)$-long sub-signals. Each sub-signal contains an independent
instance of the MRE signal, i.e., it is a triple $(s, p, \Delta)$ devised according to the rules in Section 3.3. Although all
sub-signals of a machine involve a same underlying set of observed functions $f_j^i$, the choices of their $p$ parts are
independent (i.e., they encode information of different parts of a same set of functions). The server then collects all the
sub-signals and follows a rule similar to MRE.
Proof of optimality: With a small modification of Lemma 3 and keeping the rest of the proof unchanged, it is not difficult
to show that the above idea of dividing each signal into $b/(d\log mn)$ sub-signals has an effect equivalent to multiplying
the number of machines by $b/(d\log mn)$. Therefore the expected error of the modified algorithm would be equal to
the expression in Corollary 1 with $m$ replaced with $mb/(d\log mn)$, that is $\mathbb{E}\big[\|\hat{\theta} - \theta^*\|\big] = \tilde{O}\big((mb)^{\frac{1}{\max(d,2)}} n^{\frac{1}{2}}\big)$. This
matches the lower bound in Theorem 1 of [Salehkaleybar et al. 2019] up to a polylogarithmic factor, and is thereby
order optimal.

**Minor comments: a) There are missing ellipsis in Fig 1. b) Corollary 1 seems to be stated in the wrong direction.**
**c) It would be nice to include $d$, the dimensionality in the communication budget or state the dependence clearly**
**at the beginning.** – a) Modified. b) Corrected. c) Please refer to the response to the first question of Reviewer 1.

[Meta-Review · NeurIPS 2019]

This paper provides exactly what the title suggests. Theoretically, this is a solid contribution and would be of interest to researchers in statistical learning. The reviewers felt that the author response clarified most of their technical questions. However, a number of improvements could be made to the manuscript, particularly in the related work, some additional experimental validation, and a more expanded discussion of the implications of the work in terms of network resources. In the end, the story here is clean, compelling, and has some novel insight for a problem setting that has become and will remain important in the coming years.